

# Predicting the host of influenza viruses based on the word vector

Beibei Xu[1], Zhiying Tan[1], Kenli Li[1], Taijiao Jiang[2,3] and Yousong Peng[4]

[1] College of Computer Science and Electronic Engineering, Hunan University, Changsha, China
[2] Center of System Medicine, Institute of Basic Medical Sciences, Chinese Academy of Medical Sciences & Peking Union Medical College, Beijing, China
[3] Suzhou Institute of Systems Medicine, Suzhou, China
[4] College of Biology, Hunan University, Changsha, China

## ABSTRACT

Newly emerging influenza viruses continue to threaten public health. A rapid determination of the host range of newly discovered influenza viruses would assist in early assessment of their risk. Here, we attempted to predict the host of influenza viruses using the Support Vector Machine (SVM) classifier based on the word vector, a new representation and feature extraction method for biological sequences. The results show that the length of the word within the word vector, the sequence type (DNA or protein) and the species from which the sequences were derived for generating the word vector all influence the performance of models in predicting the host of influenza viruses. In nearly all cases, the models built on the surface proteins hemagglutinin (HA) and neuraminidase (NA) (or their genes) produced better results than internal influenza proteins (or their genes). The best performance was achieved when the model was built on the HA gene based on word vectors (words of three-letters long) generated from DNA sequences of the influenza virus. This results in accuracies of 99.7% for avian, 96.9% for human and 90.6% for swine influenza viruses. Compared to the method of sequence homology best-hit searches using the Basic Local Alignment Search Tool (BLAST), the word vector-based models still need further improvements in predicting the host of influenza A viruses.

## INTRODUCTION

The influenza virus is a negative-sense, single-stranded, segmented RNA virus. Its genome is composed of eight segments and mainly encodes twelve proteins, including two surface proteins HA and NA, and ten internal proteins PB2, PB1, PA, NP, M1, M2, NS1, NS2, PA-X and PB1-F2. Influenza viruses could be mainly separated into types A, B and C, while type A could be further separated into subtypes according to the HA and NA proteins, such as H3N2, H1N1, H5N1, and so on (*Taubenberger & Kash, 2010*). Type B and C influenza viruses mainly infect humans, whereas type A can infect a wide range of species, such as birds (poultry) and mammals (pigs, bats) including humans (*Webster et al., 1992*). Among them, avian, human and swine influenza viruses are most commonly observed, and cause large health and economic loss to human society. In recent years, human infections by what were considered typical avian and swine strains have become more common, for instance

Corresponding authors
Taijiao Jiang,
taijiao@ibms.pumc.edu.cn
Yousong Peng, pys2013@hnu.edu.cn

infections due to influenza H7N9 and H5N8 (*Peiris et al., 2016*; *Su et al., 2015*). Rapid determination of the host range of a given influenza virus could assist in early evaluation of the potential risk of emerging subtypes.

Avian influenza virus is considered to be the evolutionary ancestor of all other influenza viruses (*Webster et al., 1992*). When an avian influenza virus infects a different host species and is able to spread within this new host, mutations rapidly accumulate as the viral population adapts to this host. This situation has been observed in human and swine influenza viruses. Several molecular markers for human or other influenza viruses have been identified at the amino acid level, either experimentally or by means of computational analyses (*Chen et al., 2006*; *Finkelstein et al., 2007*; *Kim et al., 2010*; *Tamuri et al., 2009*). For example, a Lysine in position 627 of internal protein PB2 tends to be favored in human strains, while avian strains usually have a Glutamic acid at this position (*Finkelstein et al., 2007*; *Kim et al., 2010*). Several studies further attempted to classify the host of influenza virus by machine learning methods (*ElHefnawi & Sherif, 2014*; *Sherif, Kadah & El-Hefnawi, 2011*; *Attaluri, Chen & Lu, 2010*). For example, *Attaluri, Chen & Lu (2010)* integrated multiple machine learning techniques to predict the host of influenza A viruses and achieved accuracies ranging from 0.84 to 0.98. However, most of these studies either used a selection of HA subtypes only, or a relatively small dataset, ignoring the real, extensive genetic diversity of influenza viruses.

In this work, we used the largest influenza dataset known to date, including 163,666 unique DNA and 150,947 unique protein sequences, to predict the host of influenza viruses based on the nucleotide and amino acid word vector. The word vector is a new representation and feature extraction method (*Mikolov et al., 2013*), which was originally developed and used in natural language processing. It was first applied to biological research in 2015 (*Asgari & Mofrad, 2015*) and proved to be useful for protein family classification and disordered protein prediction. Here, we applied word vectors to predict the host of influenza viruses and achieved an overall accuracy of 0.97, which further demonstrated its strength in biological sequence representation.

## MATERIALS AND METHODS

### Overview of this work

Figure 1 shows the workflow of this work. Firstly, all protein and DNA sequences of influenza viruses (denoted as influenza protein and DNA dataset respectively) and all non-redundant protein sequences of all species (denoted as SwissProt dataset) were collected, which were used to generate word vectors by the tool word2vec. Then, all non-redundant protein and DNA sequences of influenza A viruses with known host (avian, human and swine) were transformed into protein and DNA vectors based on word vectors. Finally, the Support Vector Machine (SVM) models were built for classifying influenza viruses of avian/human, avian/swine and human/swine based on protein or DNA vectors. A voting strategy was used to determine the predicted host for the influenza virus.

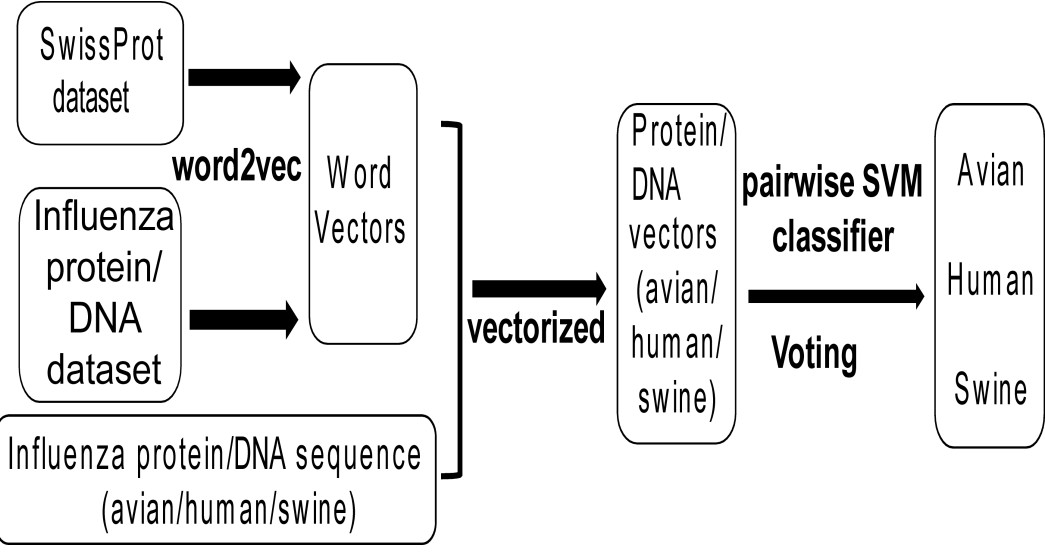

**Figure 1** **The workflow of the methodological approach used.** For explanation see the 'Materials and Methods' section.

**Table 1** **The number of non-redundant sequences used in this study for each protein of avian, human and swine influenza A viruses.**

| Protein | Avian | Human | Swine | Protein | Avian | Human | Swine |
|---------|-------|-------|-------|---------|-------|-------|-------|
| HA | 15,328 | 17,872 | 7,132 | M1 | 1,717 | 1,134 | 1,011 |
| NA | 10,295 | 9,834 | 4,802 | M2 | 2,360 | 1,456 | 1,431 |
| PB2 | 8,134 | 4,363 | 2,386 | NS1 | 5,364 | 3,128 | 2,076 |
| PB1 | 7,323 | 4,026 | 2,343 | NS2 | 2,231 | 1,000 | 995 |
| PA | 7,925 | 4,173 | 2,461 | PB1-F2 | 4,151 | 1,180 | 9,34 |
| NP | 4,816 | 2,261 | 1,929 | PA-X | 1,716 | 738 | 922 |

## Datasets

The SwissProt dataset were derived from the Swiss-Prot database on *UniProt (2016)* on April 11th, 2016. It contains 550,740 protein sequences with a length ranging from 11 to 35,213 amino acids (aa).

For the influenza DNA dataset, the nucleotide (nt) sequences for eight genes, including HA and NA as well as internal protein genes PB2, PB1, PA, NP, MP and NS, of influenza A viruses with known host (avian, human and swine) were extracted from the database of Influenza Virus Resources (*Bao et al., 2008*) on April 26th, 2016. At the same time, the amino acid sequences of the 12 proteins (HA and NA, and the ten internal proteins) were extracted to produce an influenza protein dataset. In total, 385,788 DNA and 607,327 protein sequences were collected. To reduce the computational cost, redundancy was removed at 100% level with the help of the software package cd-hit (*Li & Godzik, 2006*). This step maintained 163,666 unique DNA and 150,947 unique protein sequences for analysis. The number of protein sequences used in this study for each protein of avian, human and swine influenza viruses is shown in Table 1; the number of genes is included in Table S1. These datasets are much larger than those used in previous studies.

### Word vector generation and protein sequence vectorization

The tool word2vec is a software package developed for producing word embeddings (*Mikolov et al., 2013*). It takes a large corpus of text (here, it refers to large number of protein or DNA sequences which were separated into words) as its input and outputs a vector space of several hundred dimensions. Each unique word in the text is assigned a corresponding vector in the space, during which they are closely located to other words that share a common context. Here, the tool word2vec was used to generate the word vectors of 200 dimensions using the SwissProt dataset and the influenza datasets respectively. The skip-gram model and hierarchical softmax algorithm were used in the word2vec, with other parameters in default values. The word vectors with words of two to four amino acids (or nucleotides) long were all generated in the same way.

The vectorization of protein (or DNA) sequences was adapted from Asgari and Mofrad's work (*Asgari & Mofrad, 2015*). Firstly, each protein (or DNA) sequence was separated into overlapping words of N (2∼4) amino acids (or nucleotides). Then, the word vectors for all these words were summed up and averaged, which led to the protein (or DNA) vectors of 200-dimensions for the protein (or DNA) sequences.

### Predicting the host for the influenza virus with SVM

The SVM models for predicting the host of influenza viruses were built using functions of svmtrain() and svmclassify() (*Chang & Lin, 2011*) in MATLAB R2014b. The Gaussian Radial Basis Function kernel "rbf" with default parameters was used for the SVM models. Three SVM models were built to discriminate the influenza viruses of avian and human, avian and swine, human and swine based on word vectors. A simple voting strategy was used to determine the final prediction for the host of influenza viruses. Ten-fold cross-validations were used to evaluate the performance of the SVM models.

### Predicting the host of influenza viruses with sequence homology search

The method of the Profile hidden Markov model (HMM) through the package of HMMER3 (*Eddy, 2010*), and the method of the Basic Local Alignment Search Tool (BLAST) through the package of BLAST+ (*Altschul et al., 1990*), were used for inferring the host of influenza A viruses based on homologies of protein or DNA sequences. For each gene or protein, 75% of protein (or DNA) sequences were randomly selected for building the library (for BLAST) or profile (for HMM), while the remaining protein (or DNA) sequences were used to test through the best hit search.

## RESULTS

### Predict the host of influenza viruses based on word vectors derived from influenza protein dataset

We firstly attempted to predict the host of influenza A viruses based on word vectors derived from influenza protein dataset. Figure 2 shows that the SVM models built on the surface proteins HA and NA, and on the internal proteins PB2, PB1, PA and NP, performed much better than those built on other internal proteins did (including M1, M2, NS1, NS2, PB1-F2

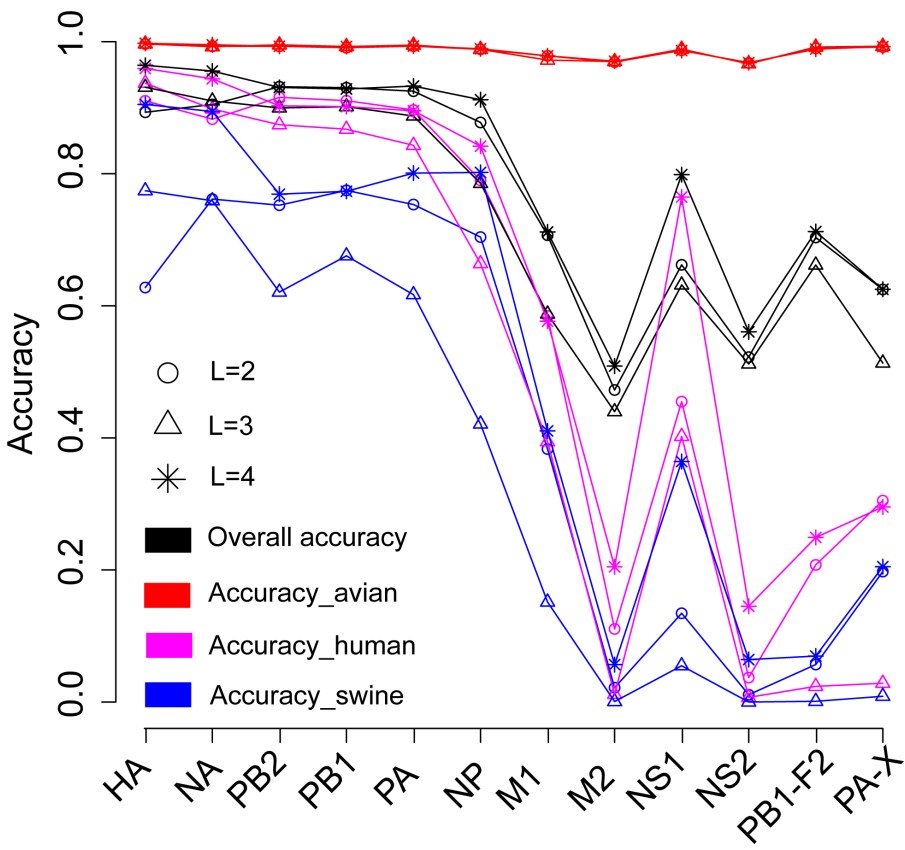

**Figure 2  Performances for the models based on the word vector with words of two to four letters long (shown in circle, triangle and star, respectively) derived from the influenza protein dataset.** The overall prediction accuracy is shown in black, while red, purple and blue lines represent the accuracies for avian, human and swine influenza viruses, respectively. The accuracies are averaged in ten-fold cross-validations.

and PA-X). The overall accuracies ranged from 0.79 ∼0.96 (summarized in Table S2). The length of word in the word vector has a significant influence on the model's performances: the models based on four-letters words performed best for all proteins. Further analyses on the model's performance by host show that all models predict most accurately for the avian influenza virus, with accuracies ranging from 0.97 ∼1. For human and swine influenza viruses, the models achieved accuracies of approximately 0.9 for HA, NA, PB2, PB1, PA and NP, while for the other proteins performance was rather poor.

## Predict the host of influenza viruses based on word vectors derived from the SwissProt dataset

We next investigated the influence of species of protein sequence, which were used to generate the word vector, on the prediction of the host of influenza viruses. The SwissProt dataset included protein sequences of virus, bacteria, fungi, plant, animal, and so on. In theory, the word vectors derived from the influenza protein dataset should reflect more accurately the influenza virus than those derived from the SwissProt dataset. Figure 3 shows the overall accuracies for the SVM models based on two kinds of word vectors with words of two to four amino acids long. The models based on two kinds of word vectors achieved
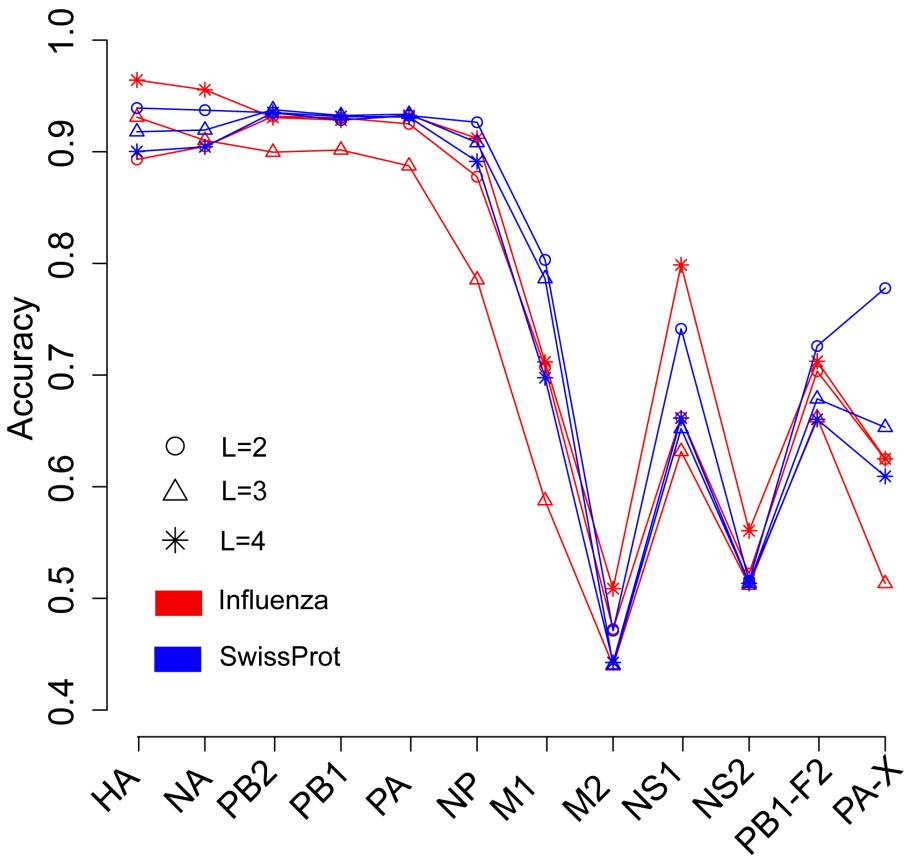

**Figure 3** Comparison of overall accuracies for the models based on word vectors with words of two to four letters long (shown in circle, triangle and star, respectively) derived from the influenza protein dataset (red line) and SwissProt dataset (blue line). The accuracies were averaged in ten-fold cross-validations.

comparable performances. For the word vector with words of two to three letters long, the models based on word vectors derived from the SwissProt dataset even outperformed those based on word vectors derived from the influenza protein dataset. However, the best performance (overall accuracy greater than 0.96) was achieved on the model built on the HA protein, which used word vectors with words of four-letters long derived from the influenza protein dataset (Tables S2 and S3). Besides, accuracies of models based on word vectors derived from the SwissProt dataset decreased as the length of the word in the word vector, while the opposite trend was observed for models based on those derived from the influenza protein dataset (Fig. 3).

## Predict the host of influenza viruses based on word vectors derived from influenza DNA dataset

We then continued to investigate the influence of data type, DNA or protein, on predicting the host of influenza viruses based on the word vector. The influenza DNA dataset were used to generate word vectors with words of two to four nucleotides long. Figure 4 shows that in most cases, models based on word vectors derived from DNA sequences outperformed
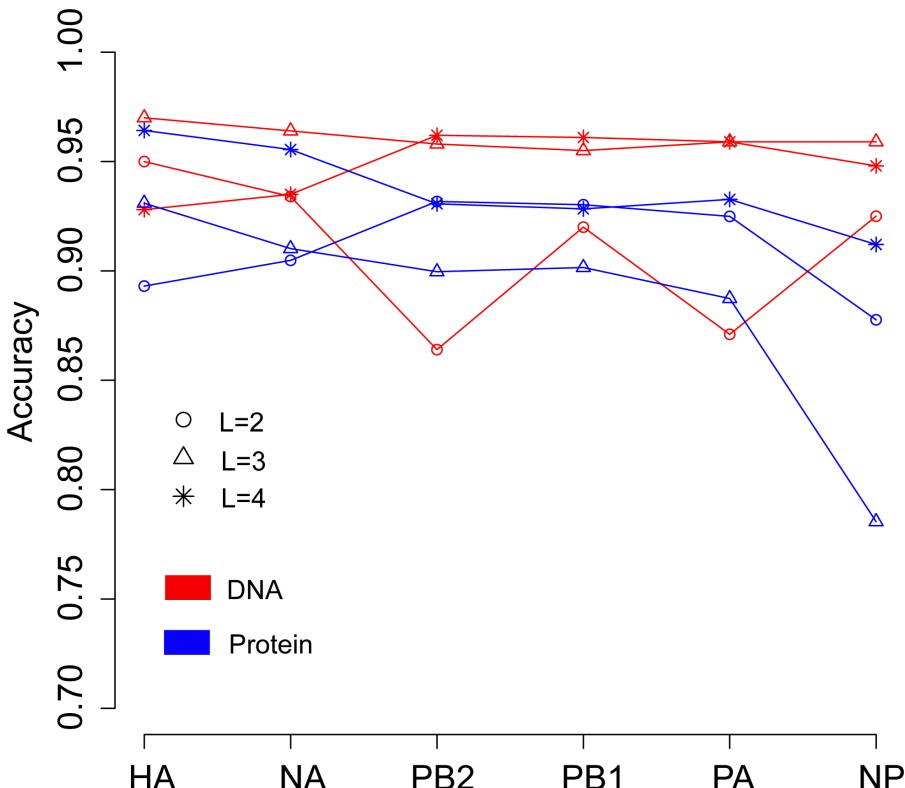

**Figure 4  Comparison of overall accuracies for the models based on word vectors with words of two to four letters long (shown in circle, triangle and star, respectively) derived from the influenza DNA dataset (red line) and influenza protein dataset (blue line).** The accuracies are averaged in ten-fold cross-validations.

those based on word vectors derived from protein sequences. Excellent performance was obtained with a DNA word length of three, with overall accuracies greater than 0.95 for most genes (Table S4). As before, the best performance was achieved on the model based on word vectors of HA gene, with an overall prediction accuracy of 0.97. More specifically, the prediction accuracy for avian, human and swine influenza viruses equaled 0.997, 0.969 and 0.906, respectively (Table S4).

## Predict the host of influenza viruses based on sequence homology search

Sequence homology search through the methods of BLAST and HMM can also be used for inferring the host of viruses. Here, we tested both methods in inferring the host of influenza A viruses based on both protein and DNA sequences. As shown in Table 2, when using the protein sequence, the method based on word vectors outperformed those based on HMM and BLAST, at least for HA, NA, and the proteins PB2, PB1, PA, NP and M2. For the other internal proteins (including the short proteins) the reverse was the case. For the methods based on sequence homology search, BLAST performed slightly better than HMM did, especially for HA and NA. Surprisingly, when using the DNA sequence, the method based

**Table 2 Comparison of methods for predicting the host of influenza A viruses based on the word vector and based on sequence homology best-hit searches using protein sequences.** The table listed the overall accuracies for predicting the host of influenza A viruses. For the method based on word vectors, the optimal model for individual proteins was used.

| Protein | Methods for predicting the host of influenza A virus | | |
|---|---|---|---|
| | Word vector | BLAST | HMM |
| HA | 0.964 | 0.950 | 0.676 |
| NA | 0.955 | 0.914 | 0.593 |
| PB2 | 0.931 | 0.892 | 0.885 |
| PB1 | 0.928 | 0.898 | 0.798 |
| PA | 0.933 | 0.917 | 0.822 |
| NP | 0.912 | 0.837 | 0.830 |
| M1 | 0.712 | 0.676 | 0.672 |
| M2 | 0.509 | 0.807 | 0.867 |
| NS1 | 0.799 | 0.864 | 0.895 |
| NS2 | 0.561 | 0.748 | 0.866 |
| PB1-F2 | 0.712 | 0.952 | 0.955 |
| PA-X | 0.625 | 0.896 | 0.730 |

on BLAST outperformed all the other methods for nearly all genes (Table S5). It achieved an overall accuracy of 0.979 on the HA gene, which is greater than that of all the other models tested here.

## DISCUSSION

This work investigated the prediction of the host of influenza A viruses based on word vectors. For all genes or proteins, the predictions for avian influenza viruses were more accurate than for human or swine influenza viruses. This may partly be caused by occasional infections of humans or swine by what actually were avian influenza viruses (*Beigel et al., 2005*; *Claas et al., 1998*), which may have weakened any host-specific signal in the non-avian hosts.

The surface proteins HA and NA were observed to be better discriminators for the host of influenza viruses than the internal viral proteins. This is most likely related to the selective pressure posed by the host: since HA and NA are the main antigens of influenza virus they will be recognized by the immune system of the host (*Couch & Kasel, 1983*). Therefore, these proteins have to mutate rapidly to maintain a stable population in a new host, a mechanism of host adaptation that will lead to divergence of lineages. Surprisingly, the results obtained with internal proteins PB2, PB1, PA and NP were comparable to those of the surface proteins. These proteins constitute the RNA polymerase complex of the virus, which is responsible for RNA replication and is thus directly responsible for the survivor ability of the virus (*Te Velthuis & Fodor, 2016*). All viral proteins are translated by the host, but these proteins are most important for rapid viral reproduction, thus they will also adapt rapidly to the new host (*Mehle & Doudna, 2009*).

As it is known, the HA proteins diverge most among all the proteins of influenza A viruses, which results in 18 HA subtypes reported until now (*Tong et al., 2013*). The best

performance achieved on the model of HA protein suggests that the word vector may capture the intrinsic difference between the influenza virus of different hosts, irrespective of the HA subtypes.

The word vectors could be generated with protein or DNA sequences of any species. In theory, the word vector derived from the influenza protein dataset could reflect more accuracy the influenza virus than those derived from the SwissProt dataset. However, in most cases, the models based on the former performed comparably with those based on the latter (Fig. 3). This may reflect the similar principles in organizing amino acids into protein sequences in all species.

A limitation of this work is that only the word vector was used in predicting the host of influenza viruses. More features such as the amino acids composition, motif frequency and molecular markers may be integrated to improve the accuracy of models in predicting the human and swine influenza viruses. Overall, this work should be an interesting attempt in using the word vector in biological sequence representation. The models based on word vectors achieved high accuracies in predicting the host of influenza viruses, which may be helpful in influenza prevention.

### Funding

This study was supported by the National Key Plan for Scientific Research and Development of China (2016YFC1200204 and 2016YFD0500300), and the National Natural Science Foundation (31500126 and 31371338). The funders had no role in study design, data collection and analysis, decision to publish, or preparation of the manuscript.

### Grant Disclosures

The following grant information was disclosed by the authors:
National Key Plan for Scientific Research and Devel-opment of China: 2016YFC1200204, 2016YFD0500300.
National NaturalScience Foundation: 31500126, 31371338.

### Competing Interests

The authors declare there are no competing interests.

### Author Contributions

- Beibei Xu performed the experiments, analyzed the data, prepared figures and/or tables.
- Zhiying Tan performed the experiments, analyzed the data.
- Kenli Li contributed reagents/materials/analysis tools, wrote the paper, reviewed drafts of the paper.
- Taijiao Jiang and Yousong Peng conceived and designed the experiments, wrote the paper, reviewed drafts of the paper.

## Data Availability

The raw data has been supplied as a supplementary file and at Github: https://github.com/flyfrommath/predicting-host-of-influenza-with-word-vector.

## Supplemental Information

Supplemental information for this article can be found online at http://dx.doi.org/10.7717/peerj.3579#supplemental-information.

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
