# Peer review of "Predicting the host of influenza viruses based on the word vector"

_PeerJ, doi:10.7717/peerj.3579_

## Round 0.1 · original submission · Major Revisions

· Academic Editor

Major Revisions

In addition to addressing specific comment by the reviewers, I strongly recommend sharing the source codes publicly (# Note from PeerJ staff: this is actually one of our requirements, at https://peerj.com/about/policies-and-procedures/#data-materials-sharing #)

Also very importantly, as both reviewers have commented, benchmarking of the method and comparison to previously published techniques is essential.

Finally, I suggest thorough editing of the paper and avoiding informal and non-quantitative language.

Reviewer 1 ·

Basic reporting

The manuscript by Xu and co-authors presents a machine-learning approach to classification of influenza viruses by their host species. Although the topic bears scientific interest, the author’s conclusion about the potential impact of their work on influenza prevention appears to be quite far-fetched. Below I discuss some specific concerns regarding the manuscript.

Major points:
1. The authors provide no comparison of their technique to other research in the field. Given that very similar study was previously conducted by ElHefnawi, 2014, although using a different training set and achieving lower accuracy, it would of utmost interest, whether the high accuracy of present study is due to a larger training set or a better model.
2. Along the same lines, I wonder how would the presented technique compare to a naive BLAST best hit search, and what are the advantages of training a machine-learning tool.

Minor points:
1. Line 31: Influenza is a negative-*sense* RNA virus.
2. Line 41: “avian” is an adjective, “birds” is a noun. I recommend a proof-reading by a native English speaker, since this is only one of many grammatical errors in the manuscript.
3. Lines 44-45: The logic of the sentence in broken.
4. Lines 56 and 57: please spell out the amino acid names.
5. Line 115: Which parameters are meant?
6. Lines 172-173: Sentence incomplete
7. Line 206: “As *it* is known…”

Experimental design

Major points:

1. The word extraction tool word2vec should be introduced in more detail. At present, I do not understand what is the difference of using it and simply splitting the sequences in overlapping k-mers?
2. The SVM kernel and the choice of it are not specified. Additionally, the software used to perform the experiments is not specified.
3. Why there is such a difference between using UniProt and influenza proteins dataset? Are they balanced with respect to the virus host? By the way, was is UniProt data or SwissProt data? The text is inconsistent.

Validity of the findings

The findings appear to be sound to the problem stated.

Reviewer 2 ·

Basic reporting

no comment

Experimental design

no comment

Validity of the findings

The benchmarks of the word vector approach need to be conducted including previously reported methods for the classification of influenza host e.g., Hidden Markov Models and Neural Networks using the same dataset.

Additional comments

The language of the manuscript needs to be more formal and generally, needs to be improved.

I believe it is important to repeat the benchmarking (figure 2, 3, and 4) for the authors' method along with the other methods cited in the manuscript. Otherwise, there is no way to evaluate the method.

It is not clear why the authors worked with two different sets of sequences (SwissProt and influenza database).

There is no reference/link to the source code, which needs to be provided along with the manuscript or preferably publicly shared (e.g., via Github).

---

## Round 0.2 · Minor Revisions

· Academic Editor

Minor Revisions

As the reviews suggest, the language of the paper would definitely benefit from further editing, specifically in the Abstract and the newly added text.

Reviewer 1 ·

Basic reporting

I doubt that the last sentence sentence of the abstract should be there ("Overall, this work is an interesting attempt in application of the word vector in biological research and may be helpful in influenza prevention."). I think it is for the reader to decide whether this work is interesting or not. I suggest to remove it completely.

English can further be corrected, e.g. in "The method of Profile hidden Markov model (HMM) , achieved with the package of HMMER3 (Eddy 2010), and the method of Basic Local Alignment Search Tool (BLAST), achieved with the package of BLAST+ (Altschul et al. 1990), were used for inferring the host of influenza A viruses based on protein or DNA sequences." "achieved" should not be there in either position. I cannot list all mistakes of this kind, but there are still some, particularly in the newly added sections.

Additionally, in "Tomas Mikolov KC, Greg Corrado, Jeffrey Dean." Tomas, Greg and Jeffrey are first names, and the reference should be edited accordingly.

Experimental design

n/a

Validity of the findings

n/a

Additional comments

The authors seem to have adequately met my concerns about the manuscript. Although they do not report any breakthroughs, their account appears to be sound.

---

## Round 0.3 · accepted · Accept

· Academic Editor

Accept

I am sure your paper will contribute to better recognizing the hosts of influenza and possibly other viruses, as more molecular data is becoming available.